# Cocaine- and Levamisole-Induced Vasculitis: Defining the Spectrum of Autoimmune Manifestations

**DOI:** 10.3390/jcm13175116

**Published:** 2024-08-28

**Authors:** Luca Iorio, Federica Davanzo, Diego Cazzador, Marta Codirenzi, Eleonora Fiorin, Elisabetta Zanatta, Piero Nicolai, Andrea Doria, Roberto Padoan

**Affiliations:** 1Rheumatology Unit, Department of Medicine DIMED, University of Padua, 35128 Padua, Italy; 2Otorhinolaryngology Section, Department of Neuroscience DNS, University of Padua, 35128 Padua, Italy; 3Department of Cardiac, Thoracic, Vascular Sciences and Public Health, University of Padova, 35128 Padova, Italy

**Keywords:** vasculitis, cocaine, levamisole, antineutrophil cytoplasmic autoantibody, ANCA, cocaine-induced midline destructive lesion, CIMDL, cocaine-induced vasculitis, CIV, levamisole-adulterated cocaine, LAC vasculopathy/vasculitis

## Abstract

Drug-induced or associated vasculitis is a prevalent form of vasculitis that resembles primary idiopathic antineutrophil cytoplasmic autoantibody (ANCA) vasculitis (AAV). Cocaine is a diffuse psychostimulant drug and levamisole is a synthetic compound used to cut cocaine. Their abuse may result in a spectrum of autoimmune manifestations which could be categorized into three overlapping clinical pictures: cocaine-induced midline destructive lesion (CIMDL), levamisole-adulterated cocaine (LAC) vasculopathy/vasculitis, and cocaine-induced vasculitis (CIV). The mechanisms by which cocaine use leads to disorders resembling AAV are not well understood. Cocaine can cause autoimmune manifestations ranging from localized nasal lesions to systemic diseases, with neutrophils playing a key role through NETosis and ANCA development, which exacerbates immune responses and tissue damage. Diagnosing and treating these conditions becomes challenging when cocaine and levamisole abuse is not suspected, due to the differences and overlaps in clinical, diagnostic, therapeutic, and prognostic aspects compared to primary idiopathic vasculitides.

## 1. Introduction

Vasculitides are a group of rare, heterogeneous diseases that primarily affect vessel walls. They can be either primary (idiopathic) or secondary to triggers such as infections, drugs, or toxins, or may occur as part of another inflammatory disorder or cancer. Clinical manifestations and organs affected vary, depending on type and size of blood vessels involved. The Chapel Hill Consensus definitions, first published in 1994 and updated in 2012, established names and definitions for the most common forms of vasculitis based on the typical distribution of vessel involvement, categorizing them into large-vessel vasculitis, medium-vessel vasculitis, and small-vessel vasculitis [1]. Antineutrophil cytoplasmic autoantibody (ANCA) vasculitis (AAV) is a systemic autoimmune disorder characterized by small-blood-vessel necrotizing inflammation. AAV is associated with the presence of autoantibodies directed against proteinase 3 (PR3) and myeloperoxidase (MPO). The clinicopathologic variants of AAV include microscopic polyangiitis (MPA), granulomatosis with polyangiitis (GPA), eosinophilic granulomatosis with polyangiitis (EGPA), and renal-limited vasculitis [1].

Many conditions, including infections, vasculopathies, and non-inflammatory disorders like atherosclerosis, can closely mimic vasculitis. It is crucial to monitor patients diagnosed with vasculitis vigilantly, particularly those who do not respond to immunosuppressive therapy. Careful differentiation between these mimics and true vasculitis is essential to prevent misdiagnosis [2,3]. Recent studies have shown that an increasing number of drugs are linked to the development of AAV, with its clinical features closely resembling those of primary AAV [3]. To date, the drugs that are most frequently identified as causing small-vessel vasculitis are hydralazine, minocycline, propylthiouracil, cocaine, levamisole, allopurinol, sulfasalazine, D-penicillamine, and gold [4,5].

With the increasingly widespread illicit use of cocaine, a broad spectrum of local and systemic disorders related to this drug abuse is emerging, involving both cocaine and levamisole [6]. The spectrum of cocaine and levamisole-induced autoimmune manifestations could be summarized into three overlapping clinical pictures: cocaine-induced midline destructive lesion (CIMDL), levamisole-adulterated cocaine (LAC) vasculopathy/vasculitis, and cocaine-induced vasculitis (CIV).

This review offers a detailed review of diseases induced by cocaine and levamisole. It highlights their distinct pathogenesis, clinical features and treatment options, and provides guidance for clinicians in managing affected patients.

### 1.1. Cocaine

Cocaine, a highly potent and addictive drug, is the most commonly used psychostimulant in Europe, and is extracted from the leaves of the coca plant (Erythroxylum coca) [7]. Cocaine raises the amount of dopamine in the synaptic cleft of the central nervous system by stopping its presynaptic reuptake and stopping the presynaptic reabsorption of noradrenaline and serotonin [7]. This causes changes in behavior and mental health.

The 2022 United Nations Office on Drugs and Crime reported that in 2021, 22 million people worldwide had used cocaine in the previous year, a figure growing year by year [8]. In the European Union, cocaine is the second most used drug after cannabis, indicating its prevalence in drug consumption patterns. In 2021, an estimated 7500 individuals received treatment for cocaine use [9].

Snorting or sniffing cocaine crystals through the nose is the primary method of administration [10,11]. Detection of cocaine in urine can be achieved by screening for its metabolite, benzoylecgonine, which persists for 48 to 72 h post-consumption and can linger for up to 2 weeks in frequent users [12,13] Cocaine is detectable in blood and saliva for less than 48 h, in sweat for several weeks, and in hair for several months [14,15].

### 1.2. Levamisole

Levamisole is a synthetic compound derived from imidazothiazole, originally used as an anti-helminthic agent in veterinary medicine and previously employed in humans for the treatment of cancer and various autoimmune diseases, due to its immunomodulatory characteristics [14,16]. In 1999, reports of severe neutropenia led to the withdrawal of the drug from the human market [17]. Approximately 70–80% of cocaine is contaminated with levamisole, which is added to enhance its volume and stimulating properties [7,14,16]. Levamisole is used for this purpose because it has a similar white powdery texture to cocaine and cannot be detected in common impurity tests like the ‘bleach test’, a popular street test for cocaine purity. In addition, levamisole enhances the addictive effects of cocaine by acting as a nicotinic antagonist, prolonging and increasing the release of glutamate [16,17,18]. Its quantification can only be performed in specialized laboratories using liquid chromatography coupled with mass spectrometry on serum and urine samples. Additionally, levamisole is difficult to detect, due to its limited renal excretion and relatively short half-life of 5.6 h [7,19,20,21].

## 2. Pathogenesis of Cocaine- and Levamisole-Related Disorders

The mechanisms linking cocaine use to AAV remain poorly understood, and the role of autoimmunity in cocaine-induced vasculitis is still under investigation. Three distinct mechanisms can be identified in the pathogenesis, which may overlap and differ in their presence among patients. Cocaine use results in nasal trauma and vascular ischemia, directly contributing to localized nasal lesions. Additionally, cocaine triggers immune system activation through the formation of neutrophil extracellular traps (NETs), further driving inflammation and tissue damage. Finally, this process can culminate in pronounced autoimmune responses, characterized by ANCA production, which exacerbates immune reactions and can lead to systemic disease.

A summary of the proposed pathogenesis is described in Figure 1.

### 2.1. Direct Nasal Trauma and Vascular Ischemia

The pathogenesis of local nasal destructive lesions induced by cocaine is not well understood, but involves factors such as local ischemia, inflammation, trauma, recurrent nasal infections and cell apoptosis [10,22,23].

Cocaine users often experience local nasal symptoms due to direct mucosal trauma caused by the high-velocity insufflation of cocaine crystals, drug adulterants, nose-picking, and superficial necrosis. However, in some habitual users, damage extends to the underlying osseous and cartilaginous structures of the nose, leading to extensive destructive nasal lesions that often spread with a centrifugal pattern [10,24]. Factors contributing to the development of midline lesions may include local decongestant treatments, smoking habits, diabetes mellitus history and an individual predisposition to bacterial superinfections of the damaged nasal mucosa, influenced by personal nasal hygiene and antibiotic usage [10,25]. Different patterns of cocaine use, in term of frequency and duration, do not appear to correlate with the extent of the lesions [24].

Vascular ischemia is the primary mechanism driving local destructive nasal lesions attributed to direct endothelial damage, an induced prothrombotic state [26] and, most notably, cocaine’s vasoconstrictive effects, which involve the stimulation of the sympathetic nervous system and endothelial cells [27,28]. In addition, cocaine induces significant dose- and time-dependent apoptosis of nasal mucosal cells, respiratory epithelial cells, and inflammatory cells within the nasal cavity [29]. This phenomenon has been confirmed in vitro and in animal studies [30,31,32], and is also correlated with cocaine’s ability to induce the expression of genes involved in oxidative stress response and DNA damage [29,33], as well as those related to apoptosis, autophagy/lysosomal activity, tissue regeneration, cell proliferation and collagen integrity [34]. Moreover, necrosis and apoptosis, the massive vasodilatation that occurs as part of the healing process during the waning effects of cocaine, leads to frequent bleeding, scab formation and extensive crusting, which obstruct breathing and prompt their removal, causing further damage [35]. Repeated exposure extends mucosal damage also to nasal cartilaginous structures [36,37,38,39,40]. Bacterial superinfection with *Staphylococcus aureus*, documented in the nasal swabs of nearly all patients, along with cocaine-induced reduction in nasal mucociliary clearance, may further contribute to local damage [27,41].

### 2.2. Immune-System Dysregulation via Neutrophils and NETs

The exact mechanisms underlying the development of autoimmunity following the use of cocaine and levamisole remain unclear, but neutrophils are crucial contributors to immune dysregulation and are inducers of tissue damage. Recent evidence suggests that the activation of the immune cascade and the loss of self-tolerance are driven by the formation of NETs [42]. Similar to observations in systemic lupus erythematosus, aberrant NET formation and defects in NET clearance are considered significant sources of excessive circulating DNA and autoantigens [43]. Different authors have demonstrated that cocaine and levamisole can activate neutrophils and induce the formation of NETs [42,44] through a mechanism dependent on M3 muscarinic receptors and the activation of intracellular pathways, including NOX, AKT, and RAF-MEK-ERK [42,45]. Moreover, NET formation has been observed both in vitro and in skin tissue samples from patients with levamisole/cocaine-associated vasculitis [44,45].

Activated neutrophils extrude NETs containing double-stranded DNA, histones and granules. These granules include classical AAV antigens, such as MPO and PR3, as well as other proteins like human neutrophil elastase (HNE), cathepsin G and lactoferrin, against which atypical perinuclear ANCAs (p-ANCAs) are formed [46]. NETs induced by cocaine and levamisole are particularly enriched in HNE, which, together with B-cell Activating Factor (BAFF), promotes the survival and differentiation of HNE-reactive B cells. Additionally, antibodies can bind NETs, increasing their immunogenicity and further potentiating their pro-inflammatory property while protecting them from degradation, similar to mechanisms described for anti-NET antibodies in systemic lupus erythematosus [44]. Lastly, NETs, cocaine, and levamisole activate the alternative complement pathway, resulting in the deposition of complement C5b-9 complex and the formation of blood clots. This mechanism plays a direct role in thrombotic microangiopathy [26,45,47,48].

The proinflammatory environment, along with the presence of autoantibodies, leads to a feedback loop of persistent neutrophil hyperactivation and amplifies immune responses, resulting in small-blood-vessel inflammation and further tissue damage [15,48].

### 2.3. Production of Autoantibodies

The detection and prolonged persistence of autoantibodies, such as ANCAs and anti-HNEs, in blood tests of patient with cocaine- and levamisole-induced disorders, but not in unaffected cocaine abusers, suggests a role for loss of tolerance and autoreactivity in the pathogenesis of both local and systemic manifestations, although this role is not yet fully elucidated [49].

Environmental factors, possibly including levamisole and infections such as Staphylococcus Aureus, likely through by superantigens [50], may stimulate the adaptive immune response, inducing neutrophil priming and activation. This leads to the generation of NETs and release of various antigens, including MPO and PR3. In predisposed subjects, antigen-presenting cells may present ANCA antigens to T cells, produce pro-inflammatory cytokines and generate ANCAs and other autoantibodies in situ, contributing to their long-term persistence [11,51,52]. This mechanism resembles the pathogenetic events described in GPA and MPA [53]. The normalization of autoantibody titers after discontinuing cocaine and levamisole may confirm their direct effect on NETosis and ANCA development [19,49].

Conversely, anti-HNE antibodies do not seem to directly affect enzyme activity or its ability to bind to natural inhibitors [54], but they may significantly enhance the local inflammatory response to injury by promoting inflammation and necrosis [20,55,56]. This effect is achieved by disrupting the non-inflammatory clearance of apoptotic cells by macrophages. In vitro studies suggest that this mechanism may involve opsonization, which increases the production of inflammatory cytokines by phagocytosing macrophages and reduces the expression of phosphatidylserine, the recognition signal for macrophages, on the surface of pre-apoptotic cells [57].

A number of studies have shown that the presence of high titers of different ANCAs and/or the presence of anti-HNE in the blood of patients can be strong indicators of exposure to levamisole and cocaine. This “dual positivity” is rarely found in other autoimmune diseases or vasculitis, and anti-HNEs are never present in patients with GPA and MPA, nor in subjects exposed to cocaine without autoimmune manifestations [11,49,58,59]. Some studies speculated about the significance and role of anti-HNE in these individuals. Since HNE is a component of neutrophil granules, it might present a perinuclear ANCA (p-ANCA) or atypical pattern in indirect immunofluorescence (IIF). Moreover, HNE shares sequence and structural homology with PR3, which can also lead to a cytoplasmatic ANCA (c-ANCA) pattern in IIF [48,60]. The presence of the HNE antigen and anti-HNE antibodies may provide an explanation for the widespread “dual positivity” observed in these patients. In laboratories where enzyme immunoassays for HNE antigen are unavailable, the discordance between a high titer of perinuclear ANCA and a low titer in the MPO-ANCA immunoassay can be a useful indicator of cocaine and levamisole exposure [19,49,60,61,62].

## 3. Clinical Presentation of Levamisole- and Cocaine-Induced Disorders

### 3.1. Cocaine-Induced Midline Destructive Lesion (CIMDL)

Long-term insufflated cocaine use can lead to a syndrome known as CIMDL, characterized by extensive damage to the osseocartilaginous structures of the midface.

Although the prevalence of CIMDL is not known, data from the 1998 report of the United States Department of Health and Human Services shows that 4.8% of cocaine abusers are affected by an isolated nasal septum perforation, which is the most common clinical presentation of CIMDL [52].

#### 3.1.1. Clinical Features

The symptoms of CIMDL are related to the destruction of the sinonasal district. Initially, patients may experience nasal obstruction, epistaxis, hyposmia or anosmia, nasal crusting, sinusitis, severe oral–facial pain and altered sensitivity. If the lesions extend to the middle and superior turbinates, lateral nasal wall, and hard and soft palate, causing perforations, symptoms can include dysphagia, oronasal reflux, regurgitation, hypernasal speech, excessive sniffing and halitosis [10,63]. In cases where superinfections propagate the damage, some patients may develop diplopia, proptosis, pseudotumor and headache [27,63]. Patients do not present with systemic symptoms, such as fever, malaise, weight loss, arthralgia or myalgia, or with the involvement of another specific organ [63].

The most common signs, detectable by nasal endoscopy, are diffuse necrotizing ulcerative lesions, extensive crusting and nasal septal perforation. In more severe cases, external features like nasal deformity, saddle nose, loss of nasal projection, hard and/or soft palate perforations, erosion of middle and superior turbinate and sphenoid bone erosion may be observed (Figure 2 and Figure 3) [10].

Other rare manifestations include pharyngeal wall ulcerations (Figure 3) [64,65], nasolacrimal duct obstruction and destruction of the orbital bony walls, which can lead to secondary orbital cellulitis [66]. In these cases, damage could range from dehiscence of the lamina papyracea or of the orbital floor without visual symptoms to more severe conditions, such as chronic orbital inflammation, double vision, reduced visual acuity, ocular motility impairment, pseudotumor and optic neuropathy. Additionally, erosion of the anterior skull base can cause cerebrospinal fluid leak, encephalocele or acute diffuse pneumocephalus, associated with altered mental status, severe headache, anxiety, confusion, cerebritis and meningitis, or panhypopituitarism in the case of pituitary gland damage [10]. Brain damage caused by the spread of vasculitic lesions or tumor-like growths from the nose has also been reported [67,68].

Other signs related to the nasal route of administration include manifestations of facial and external structures, with external cutaneous ulcers on the nose and surrounding area, along with lip swelling and phymatoid enlargement of the nose [10].

#### 3.1.2. Histopathology

Multiple biopsy specimens, especially of the nasal mucosa, should be taken from the margins of the lesions, avoiding the necrotic center, which is unlikely to provide any diagnostic information [69].

There are no pathognomonic histopathological features for CIMDL. Histopathology may reveal a vasculitis-like granulomatous pattern, which shares many features with GPA, including micro-abscesses in vascular walls, leukocytoclastic vasculitis, mixed acute and chronic inflammatory infiltrates, perivenulitis, vascular microthrombotic changes and fibrinoid necrosis. However, granulomatous inflammation has not been described [10,27]. A distinctive feature of CIMDL might be massive apoptosis, which is not observed in healthy subjects or in patients with nasal polyposis or GPA. It can be detected using tools like the commercially available in situ terminal deoxynucleotidyl transferase-mediated dUTP-digoxygenin nick end labeling (TUNEL) cell-death detection kit [29].

#### 3.1.3. Laboratory Findings

Positive ANCA tests are often detected in CIMDL [10,27,49,54,70]. In IIF, the predominant pattern observed is p-ANCA, as confirmed in ELISA through single or dual ANCA positivity. This is attributed to the presence of anti-HNE, observed in up to 84% of CIMDL cases, or PR3-ANCA, noted in up to 57% of cases. Moreover, the presence of p-ANCA in IIF, may be associated with a low titer of anti-MPO antibodies, as observed also in CIV and LAC vasculopathy/vasculitis [10,27,49,54,70]. Additionally, PR3-ANCA with c-ANCA in IIF can be found in about 50% of ANCA-positive CIMDL patients [49,54]. Some patients may be negative in both IIF and solid-phase assays [49,54]. Studies comparing ANCA-positive and ANCA-negative CIDML patients have found no significant differences in presentation, symptoms, laboratory tests, radiological and histological findings, type of treatment and disease severity [11].

#### 3.1.4. Imaging

Magnetic resonance imaging (MRI) and computed tomography (CT) scans with and without contrast agent are used to assess the presence and extent of midline destructive lesions, though these imaging techniques lack specificity for diagnosing CIMDL. High-resolution CT scans of the head reveal cartilage and/or bone reabsorption in the sinonasal area, with the affected structures displaying a progressively centrifugal pattern over time (Figure 4). Other common findings include the opacification of the paranasal sinuses and mucoperiosteal thickening of the nasal cavity and paranasal cavities [71]. MRI with and without contrast agent, often the preferred initial choice, is used to evaluate soft-tissue erosion and mucosal inflammatory lesions, which appear hypointense on T2-weighted images with reduced or nonhomogeneous enhancement [10,69,72]. Additional MRI findings in cocaine abusers may include radiographic signs of otitis media [72] and diffuse swelling of palatine and pharyngeal tonsils, often accompanied by small fluid collections within lymphatic tissue [27].

Based on the localization pattern, Nitro and colleagues proposed a radiological classification of CIMDL. They divided the sinonasal district into four progressively less-involved areas, starting with the nasal septum and extending to the neurocranial structure. This suggests that the nasal septum is the starting point of the process, with lesions potentially spreading centrifugally across the sinonasal compartment [24].

This radiological classification is reported in Table 1.

### 3.2. Cocaine-Induced Vasculitis (CIV)

Active cocaine use can also induce systemic vasculitis. Several case reports and series have described cases that resemble the classic presentation of primary idiopathic AAV, particularly GPA, but occurring in the context of cocaine abuse [61,62,73,74,75,76,77]. Other types of CIV have also been documented, including cases of IgA vasculitis, EGPA and isolated central-nervous-system vasculitis [62,78,79,80,81,82,83].

#### 3.2.1. Clinical Features

Almost all patients experience nasal symptoms, similar to those seen in individuals with local cocaine-snorting damage or GPA [62,77,84]. Systemic involvement includes skin rashes, joint involvement, pauci-immune crescentic glomerulonephritis (GN) and other specific organ manifestations [60,62,75,85]. Additionally, renal complications such as renal infarction, thrombotic microangiopathy, and acute kidney injury from rhabdomyolysis are also associated with cocaine abuse [75].

#### 3.2.2. Histopathology

The skin biopsy reveals leukocytoclastic vasculitis, affecting small and medium-sized dermal vessels, characterized by fibrinoid necrosis of the vessel walls and a neutrophil infiltrate with nuclear debris. Direct immunofluorescence of the small vessels shows large deposits of Immunoglobulin M (IgM), along with smaller amounts of IgG, IgA and C3 [77,84]. Renal pathology can manifest in various forms, most commonly as pauci-immune necrotizing GN, but also as membranous nephropathy, focal segmental glomerulosclerosis, C3-associated GN, mesangialproliferative/IgA-associated GN, and immune complex-mediated GN [75,84].

#### 3.2.3. Laboratory Findings

The majority of patients, up to 90%, test positive for ANCAs [62]. These patients typically show positive PR3 antibodies on ELISA and p-ANCA on IIF or, less commonly, dual positivity for both PR3- and MPO-ANCA [49,84]. As mentioned earlier, this non-congruent and dual positivity is common among cocaine abusers [60,62,86]. Some studies have reported a higher prevalence of MPO positivity compared to other types of ANCAs [62,87]. A minority of CIV patients may be ANCA-negative [60,88].

### 3.3. Levamisole-Adulterated Cocaine (LAC) Vasculopathy/Vasculitis

LAC vasculopathy/vasculitis is a complex systemic syndrome that primarily affects the skin and leukocytes, where the inflammatory vascular effects of cocaine and levamisole exposure converge. The first two reports of vasculopathy/vasculitis induced by levamisole-contaminated cocaine were published in 1978 [89,90]. This condition is more commonly found among middle-aged women [21,58].

#### 3.3.1. Clinical Features

LAC causes a clinical syndrome primarily characterized by skin involvement (91% of cases) and can also affect other organs such as the nose, kidneys, lungs, liver and brain [7,16]. The cutaneous presentation includes retiform or stellate purpura, skin rash, necrotic ulcer, painful hemorrhagic blisters, and pyoderma gangrenosum-like lesions. The most common feature is retiform purpura with central necrosis and an erythematous edge, typically localized to the ears, cheeks, zygomatic arch and lower extremities, while sparing the trunk and neck. Skin involvement is usually bilateral [15,21,61,91,92,93,94,95,96]. Some authors identified four different stages of retiform purpura, beginning with livedo reticularis or racemosa and progressing to purpura without necrosis. The condition worsens through an intermediate stage characterized by purpuric macules and plaques with hemorrhagic vesicles, eventually leading to extensive confluent purpura with ulceration and necrosis [59,97,98].

Constitutional symptoms, such as arthralgias, fever, weight loss, myalgia and sweats are very common in LAC vasculopathy/vasculitis, as are midline destructive lesions due to cocaine insufflation and sinonasal involvement. Musculoskeletal involvement is also highly prevalent, with some case series reporting a prevalence of up to 90%. Arthritis, present in 30% of cases, typically manifests as acute inflammatory arthritis, though a small proportion of anti-MPO-positive patients develop chronic erosive arthritis [95]. Less frequent features include pulmonary involvement, leukoencephalopathy and GN [16,19,21,58,60,61,95]. Pulmonary involvement most commonly includes alveolar hemorrhage, but can also present as bronchitis and bronchiolitis, interstitial pneumonia, hypersensitivity pneumonitis and subcentimeter pulmonary nodules [87,99]. Leukoencephalopathy, first observed in individuals treated with levamisole, is an autoimmune reaction that leads to progressive mental-status deterioration, ataxia, and other neurological symptoms [19,100]. Renal involvement occurs in up to 10% of cases, often at disease onset, and may manifest as proteinuria and hematuria, with or without renal failure [21,58,59].

In some cases, LAC vasculopathy/vasculitis can take a fulminant course, leading to a fatal outcome due to the severity of skin lesions or the involvement of multiple organs [19,58].

#### 3.3.2. Histopathology

Cutaneous histological findings are characterized by thrombotic vasculopathy, leukocytoclastic vasculitis or a combination of both [7,16]. Superficial and deep small dermal vessels are involved and show mixed inflammatory cell infiltrates, characterized by a prominence of neutrophils and sometimes eosinophils. Fibrinoid necrosis of vascular walls is often observed and frequently extends into the adjacent perivascular connective tissue. Extravasation of red blood cells and the presence of intravascular thrombi are also common findings. Direct immunofluorescence may reveal mixed antibody (IgM, IgG and IgA) and C3 deposition in vessel walls [17,87,101]. Interestingly, more than half of the patients with vasculopathy on biopsy tested positive for antiphospholipid antibodies [58].

The most common kidney histological pattern is pauci-immune GN, which may or may not include cellular crescents and fibrinoid necrosis, similar to most primary AAVs. Membranous GN, with or without anti-phospholipase A2 receptor antibodies, is also relatively common [21,88,102].

#### 3.3.3. Laboratory Findings

Hematological abnormalities, such as leukopenia, neutropenia, agranulocytosis, hemolytic anemia and thrombocytopenia are common, occurring in about 60% of cases [7,16,19,61,94]. Agranulocytosis, characterized by an absolute neutrophil count of less than 0.5 × 10^9^ cells/L, occurs in 6 to 13% of patients currently taking levamisole. It typically develops 1–2 weeks after the initial exposure to levamisole or immediately following re-exposure [16]. The exact mechanism by which some patients develop levamisole-associated agranulocytosis and neutropenia is not fully understood, but is not a feature of cocaine-linked autoimmunity. Some studies suggest that individuals carrying the human leukocyte antigen (HLA)-B27 haplotype are more likely to develop agranulocytosis when exposed to levamisole [16,103,104,105]. Common features of levamisole-induced agranulocytosis include the presence of plasmacytoid lymphocytes in peripheral blood, increased plasma cells and megakaryocytic hyperplasia in bone marrow [105].

ANCA positivity occurs in up to 90% of cases [58,87]. LAC vasculopathy/vasculitis may be associated with p-ANCA and/or c-ANCA or both, with multiple antigenic specificities [58,87]. High titers of atypical p-ANCA, directed against HNE, cathepsin G and lactoferrin, are the most frequent pattern. Anti-MPO antibodies are typically identified at lower levels compared to those seen with p-ANCA on IIF. Other case series report that many patients with p-ANCA and/or anti-MPO also exhibit positivity for the PR3 immunoassay [16,19,54,58,60,106].

Antiphospholipid antibodies, such as lupus anti-coagulant, IgM anti-cardiolipin antibody and IgM anti-β2 glicoprotein-1 antibody, are also very common, occurring in nearly 70% of cases. Additionally, about the same percentage of patients have low complement levels (C3, C4 or both). In some cases, antinuclear antibodies (ANAs), anti-double-stranded DNA antibodies and anti-C1q antibodies are also present [16,19,58,60,107].

## 4. Differential Diagnosis

Distinguishing between the three described clinical pictures, as well as differentiating idiopathic vasculitis from other vasculitis mimickers, can be challenging. Confirming substance abuse through patient history or laboratory tests is valuable but not always feasible in routine clinical practice.

Midfacial osseocartilaginous destruction can result from various conditions beyond cocaine abuse, including autoimmune diseases, chemical exposure or trauma [10]. Microbiological and histopathological assessments help differentiate among infections (e.g., tuberculosis, tertiary syphilis, mucormycosis), malignancies (lymphoma, squamous cell carcinoma) or granulomatosis lesions (e.g., GPA, sarcoidosis and Immunoglobulin G4-related disease) [15,22,23,108,109]. Notably, Immunoglobulin G4-related disease (IgG4-RD) can involve midline structures with mass-forming lesions and nasal or palatal erosion, although this is rarely reported [110]. The specific histology of IgG4-RD aids in distinguishing it from CIMDL [15,109]. Differentiating CIMDL from GPA with limited ear–nose–throat (ENT) involvement is particularly challenging, as nasal manifestations are present in 50–90% of GPA cases and ANCA testing is not always definitive [74,111,112,113]. A histologically definitive diagnosis of GPA requires the identification of pathognomonic lesions, such as stromal granulomas with giant cells and deep necrosis, which occur in approximately 50% of GPA patients with nasal involvement [27,63,69,114,115]. While up to 83% of GPA patients test positive for ANCA, this percentage is lower in localized sinonasal disease [52,116]. Most GPA patients have c-ANCA with anti-PR3 positivity, whereas a minority have p-ANCA with anti-MPO positivity or are ANCA-negative [69]. A c-ANCA/anti-PR3 pairing suggests GPA, while a positive p-ANCA, and negative anti-MPO, with or without anti-PR3, warrants further testing for anti-HNE. A positive anti-HNE or an ANCA directed against multiple antigens may indicate a cocaine-induced etiology [11,69]. Midline facial destruction is typically more severe in CIMDL, with centrifugal involvement and reduced or non-uniform enhancement in the nasal septum and turbinate mucosa being more frequent in CIMDL than in GPA [27]. Trimarchi and colleagues suggest that palate perforation, which is reported only in CIMDL patients, could be a useful clinical marker. [22]. Investigating cocaine use should be considered at the initial presentation of isolated midline destructive lesions or when treatment failure in GPA occurs [117].

CIMDL can be distinguished from CIV by its lack of systemic involvement. Differentiating CIV from idiopathic autoimmune vasculitis, however, is more challenging [10,112]. Clinical manifestations and histological findings often do not exclude idiopathic forms of vasculitis [16,62]. Dual antibody positivity, high-titer atypical p-ANCA, or the presence of anti-HNE are useful “red flags”. The resolution of vasculitis symptoms upon cessation of cocaine use, along with the persistence of symptoms despite standard treatment, should raise suspicion of cocaine abuse [14,61,118].

Differentiating LAC vasculopathy/vasculitis from idiopathic AAV can be difficult, but some differences exist. Clinically, purpura is more prevalent and severe in LAC vasculopathy, often requiring surgical management, unlike in AAV [7,58,59,119]. Additionally, the variability and location of skin involvement differ, particularly with facial lesions, which typically do not require surgical intervention in AAV [58,59]. Biologically, the presence of dual-positive ANCA tests and/or antiphospholipid antibody tests in AAV patients can be major “red flags” for levamisole abuse [58]. Furthermore, LAC vasculopathy frequently causes neutropenia, which is uncommon in AAV [19,105]. Lastly, relapses of LAC vasculopathy/vasculitis are closely linked to re-exposure to levamisole, unlike AAV, where relapses occur unpredictably [58].

For a comprehensive overview of the hallmarks and distinctions among CIMDL, LAC vasculopathy/vasculitis, and CIV, please refer to Table 2 and Figure 5.

## 5. Treatment

No standardized therapeutic guidelines have been established [62]. The cornerstone of treatment remains the discontinuation of cocaine use, as the effectiveness of any therapeutic approach depends heavily on sustained abstinence from cocaine and/or levamisole [58,62,69]. In the initial stages, all three conditions may gradually normalize with the cessation of cocaine and levamisole use, highlighting the crucial role of substance cessation in the management [58,62,72]. Antibodies typically revert to negative within 2 to 14 months after discontinuation of cocaine use [11,60,62,120].

In cases of local nasal destructive lesions, treatment options include medical therapy and prosthetic or reconstructive surgery, both of which require stable lesions and a period of abstinence. Some authors recommend confirming cocaine abstinence with toxicological tests [121,122,123] and suggest a full rehabilitation period of 6 to 24 months before surgery [38,72,120], based on the high incidence of relapses within the first 12 months [124]. Conservative local treatments, such as debridement of necrotic tissues and crusts, regular saline douches, and local or systemic application of antibiotic therapy are recommended to slow lesion progression, though they cannot reverse established damage [72]. Intranasal steroids may help control nasal symptoms and infections [125]. Prosthetic closure of oronasal fistulas, septal perforation, and skull-base defects are common procedures for prosthetic surgery [35,72,121,126,127,128,129]. Obturator prostheses for palate defects can prevent oronasal reflux and serve as an early, non-invasive treatment option for patients beginning their abstinence period [121]. Reconstructive surgery for mucosal or cutaneous defects should be postponed until lesions are stable, especially for aesthetic procedures such as rhinoplasty and nasal cutaneous fistulae closure [39,72,130,131].

The first line of treatment for early-stage skin manifestations of LAC vasculopathy/vasculitis includes supportive measures, such as proper wound care, antibiotics for infected lesions and analgesics [7,48,92,101,119]. Some patients may require surgical management, including debridement and skin allo- or auto-grafting of necrotic lesions [106,119,132]. Topical corticosteroids are effective for non-necrotic lesions, such as pyoderma gangrenosum-like lesions and purpuric skin rash [96,133]. In cases of agranulocytosis, empiric antibiotic treatment and granulocyte colony-stimulating factor are recommended if fever or infection is present, as neutrophil counts generally recover within a few days after discontinuation of levamisole [16,105]. The effectiveness of anticoagulation, antiplatelet and peripheral vasodilating therapies remains unclear [7,21,59,95].

The role of immunosuppressive therapy is controversial [52,58,60,62]. Several immunosuppressive agents have been mentioned in the literature, including GCs, methotrexate (MTX), thalidomide (THD), azathioprine (AZA), cyclophosphamide (CYC), mycophenolate mofetil (MMF), and rituximab (RTX) [52,58,59,62].

Immunosuppressive therapy is generally not recommended for CIMDL. However, some authors suggest considering immunosuppressors if there is no improvement after at least 3 months of cocaine cessation or in patients with significant inflammatory disease, with GCs, MTX, or CYC as reported options [52].

Patients with CIV are treated with immunosuppressive therapy, which may include GCs, and other agents such as MTX, AZA, MMF, CYC, and RTX [62,77]. Treatment schedules and drug dosages generally align with international guidelines for idiopathic AAV, with the exception of avoiding GC pulses and using reduced dosages of GCs [77]. Although data on treatment efficacy are scarce, it is clear that cessation of cocaine abuse is a prerequisite for achieving remission [62].

The most employed immunosuppressive treatment for LAC vasculopathy/vasculitis is GC, typically starting at a dosage of 1 mg per kilogram of body weight per day, with relatively brief treatment durations. Only a minority of patients in the literature have been treated with immunosuppressive agents, including MTX, AZA, THD, CYC and RTX [58,59]. There is currently no convincing evidence that immunosuppressive therapy modifies the clinical course, but it may be reserved for patients who do not respond to supportive therapy. Patients with LAC vasculopathy/vasculitis with predominant systemic involvement, systemic inflammation, or with histologically proven vasculitis (not necrosis), particularly GN, have shown the best response to systemic GCs and immunosuppressive treatments [7,48,59,61,88,134]. Systemic GCs, with or without associated immunosuppressive treatment and plasma exchange, are mandatory in cases of leukoencephalopathy involvement [100].

In light of these considerations, it is imperative that treatment paradigms be tailored to reflect the specific severity and degree of organ involvement in each individual case.

## 6. Conclusions

The intricate relationship between cocaine use, particularly when adulterated with levamisole, and autoimmune vasculitis presents a multifaceted clinical challenge. The abuse of these substances may result in a spectrum of autoimmune manifestations in predisposed individuals, ranging from localized to systemic effects, which may partially overlap. The pathogenesis of these disorders, while not fully understood, involves complex mechanisms with immune dysregulation playing a pivotal role. These entities exhibit overlapping clinical features with primary idiopathic AAV, but are distinguished by their association with drug use.

Distinguishing between cocaine-associated vasculitis and primary vasculitis poses a significant diagnostic challenge, requiring a comprehensive approach involving detailed patient history, thorough clinical examination, and a combination of laboratory and histopathological investigations. Identifying substance abuse is crucial, yet often difficult, underscoring the need for heightened clinical awareness and a multidisciplinary approach to management.

Ultimately, these findings emphasize the critical importance of recognizing the potential for cocaine and levamisole to induce significant autoimmune pathology. To date, most of our knowledge on cocaine- and levamisole-induced vasculitis has been derived from observational studies, primarily case reports. Experimental and prospective research is needed to better understand the pathogenesis of these conditions and to develop targeted therapeutic strategies that can mitigate the harmful effects of these substances on the immune system and overall health.

## Figures and Tables

**Figure 1 jcm-13-05116-f001:**
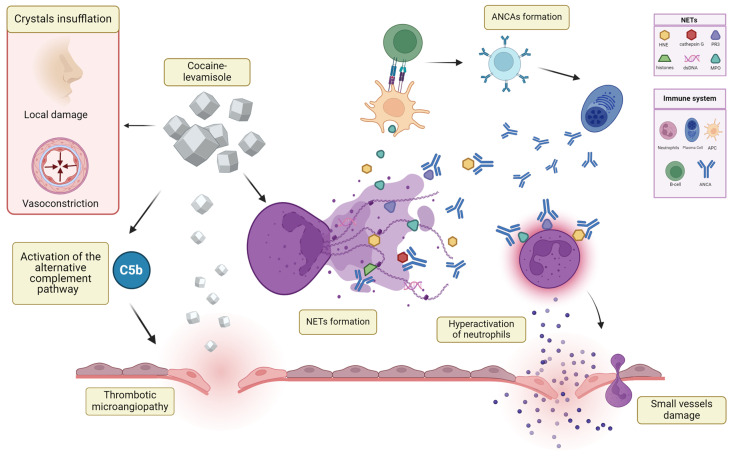
A model of the immunopathogenesis underlying damage to local nasal structures and autoimmune reactions in cocaine and levamisole-induced vasculitis (Created with www.app.biorender.com accessed on 25 May 2024). ANCAs: anti-neutrophils cytoplasmatic antibodies; APC: antigen-presenting cell; MPO: myeloperoxidase; NETs: neutrophil extracellular traps; PR3: proteinase 3.

**Figure 2 jcm-13-05116-f002:**
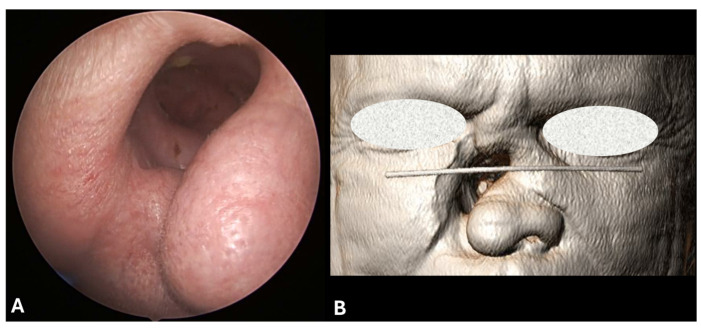
(**A**) External endoscopic view of a right paralatero–nasal cutaneous fistula in a patient with history of cocaine abuse. (**B**) The same cutaneous defect is visible in a 3D-rendering based on CT scan.

**Figure 3 jcm-13-05116-f003:**
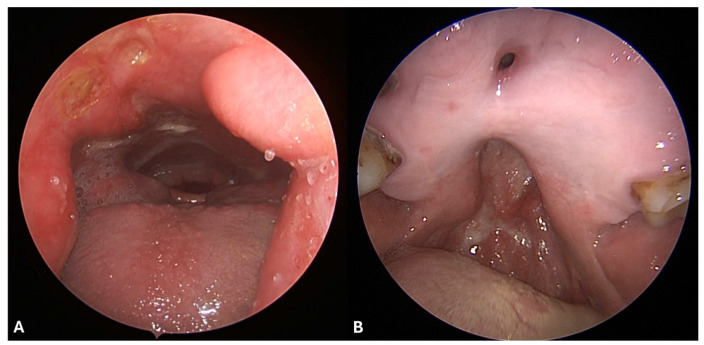
(**A**) Nasal endoscopy with a 30-degree endoscope shows a large hard- and soft-palate defect with evidence of oral cavity and laryngopharyngeal structures such as oral tongue, base of the tongue with hypertrophic lingual tonsils, free margin of the epiglottis, laryngeal vestibule, arytenoids, and retrocricoid space. A cluster of three ulcerated lesions of the posterior wall of the nasopharynx is visible. (**B**) Endoscopic view with a 30-degree endoscope from the oral cavity in the same patient revealed an anterior hard-palate fistula, and a larger hard- and soft-palate defect.

**Figure 4 jcm-13-05116-f004:**
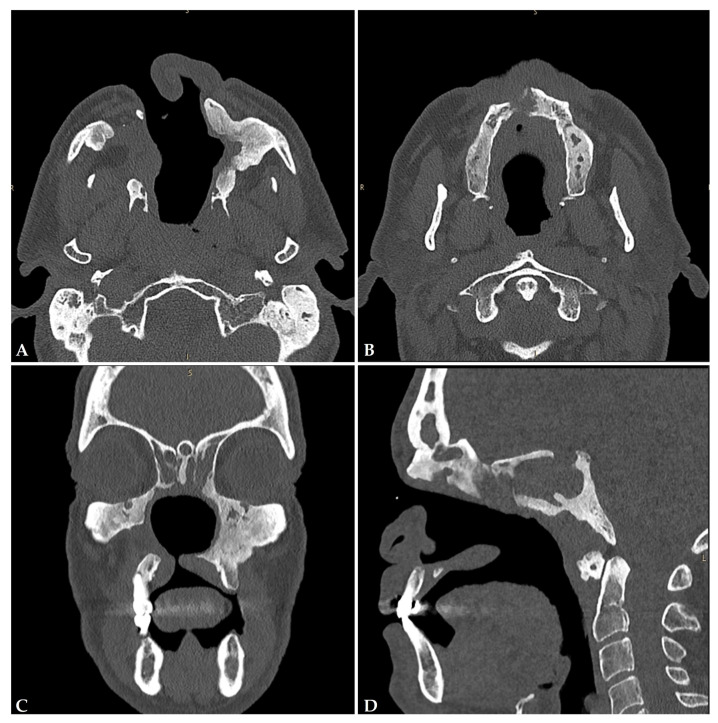
CT scan of a patient with a history of cocaine abuse, showing in the axial scans (**A**,**B**) a complete erosion of the nasal septum and bony limits of the paranasal sinuses, with right paralatero–nasal fistula and hyperostosis of the residual left maxillary-sinus bony walls. This is more evident in the coronal scan (**C**), where an oronasal fistulation of the hard palate is visible. The ethmoidal cells are obliterated, as well as the frontal and sphenoid sinuses (**D**).

**Figure 5 jcm-13-05116-f005:**
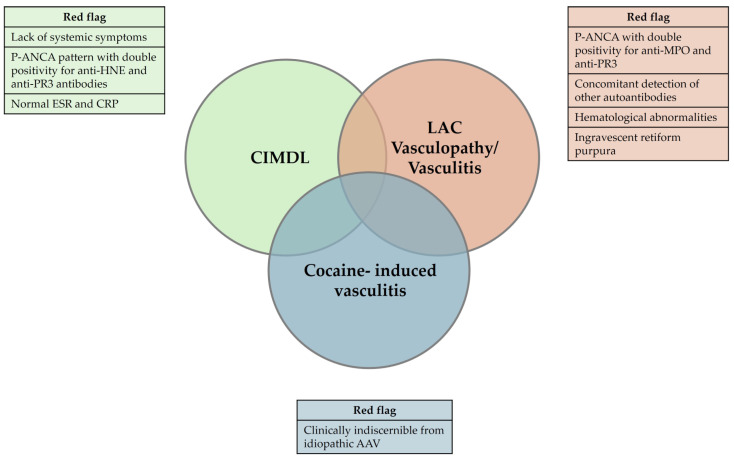
The abuse of cocaine may result in a “spectrum of autoimmune manifestations” in predisposed individuals, ranging from localized to systemic effects. These manifestations, which may partially overlap, can ultimately be classified into three clinical entities: cocaine-induced midline destructive lesion (CIMDL), levamisole-adulterated cocaine (LAC) vasculopathy/vasculitis, and cocaine-induced vasculitis (CIV). AAV: ANCA-associated vasculitis; ANCA: anti-neutrophil cytoplasmatic antibody; CRP: C-reactive protein; ESR: erythrocyte sedimentation rate; HNE: human neutrophilic elastase; MPO: myeloperoxidase; p-ANCA: perinuclear-ANCA, PR3 proteinase 3.

**Table 1 jcm-13-05116-t001:** Radiological classification of CIMDL according to Nitro et al. [24].

Classification Grade	Localization	Patients in the Nitro et al. [24] Systematic Review
1 (middle)	Nasal septum	126 (99.2%)
2 A (inferior third of the sinonasal complex)	Grade 1 + inferolateral district (inferior turbinate and Maxillary-sinus medial wall, nasolacrimal duct)	75 (59%)
2 B (inferior third of the sinonasal complex)	Grade 1 + palate (nasal floor)	38 (29.9%)
3 (middle third of the sinonasal complex)	Grade 2 + ethmoid bone, middle turbinate and superior turbinate	29 (22.8%)
4 (neurocranial structures)	Grade 3 + neurocranium (lamina papyracea, orbit base or skull base)	10 (7.9%)

**Table 2 jcm-13-05116-t002:** Differentials in clinical, laboratory, histological and treatment characteristics of CIMDL, LAC vasculopathy/vasculitis and cocaine-induced vasculitis.

	CIMDLs	LAC Vasculopathy/Vasculitis	CIV
**Disease** **localizations**	ENT (midline structures)	Local (cutaneous) and systemic	Multi-organ and systemic
**Disease manifestations**	Diffuse necrotizing ulcerative lesions, nasal crusting, nasal-septum perforation, palatal perforation, nasal deformity	Retiform purpura, constitutional symptoms, midline destructive lesions, glomerulonephritis, pulmonary involvement, leukoencephalopathy, arthritis	Skin rashes, joint involvement, pauci-immune crescentic glomerulonephritis, alveolar hemorrhage
**ANCA IIF pattern**	Perinuclear or cytoplasmatic	Atypical perinuclear	Perinuclear
**ANCA specificity**	Anti-HNE, PR3-ANCA, ANCA negative	MPO-ANCA, PR3-ANCA, anti-HNE	PR3-ANCA, MPO-ANCA, ANCA negative
**ANCA** **double positivity**	HNE and PR3	c-ANCA and p-ANCA	PR3 and p-ANCA
**Other biomarkers**	-	APL, low complement levels, ANAs, anti-dsDNA, anti-C1q, cryoglobulin	-
**Systemic** **inflammation**	Rarely	Common	Common
**Hematological** **abnormalities**	None	Common (leukopenia, neutropenia, agranulocytosis, hemolytic anemia, thrombocytopenia)	-
**Histology**	Not specific except for cell apoptosis and extensive necrosis	Thrombotic vasculopathy, leukocytoclastic vasculitis	Leukocytoclastic vasculitis, commonly pauci-immune necrotizing glomerulonephritis
**First approach**	Cessation of cocaine abuse	Cessation of cocaine abuse	Cessation of cocaine abuse
**Immunosuppressive treatment**	Rarely required, only in case of systemic inflammation	Required in case of severe systemic inflammation or life-threatening symptoms	Controversial
**Local and surgery treatment**	Conservative local treatment, prosthetic and reconstructive surgery	Supportive local therapy and plastic surgery	-

ANAs: anti-nuclear antibodies; ANCA: anti-neutrophil cytoplasmatic antibody; anti-dsDNA: anti-double-stranded DNA antibody; APL: antiphospholipid antibody; c-ANCA: cytoplasmatic-ANCA; CIMDLs: cocaine-induced midline destructive lesions; CIV: cocaine-induced vasculitis; ENT: ear, nose and throat; HNE: human neutrophil elastase; IIF: indirect immunofluorescence; LAC: levamisole-adulterated cocaine; MPO: myeloperoxidase; p-ANCA: perinuclear-ANCA; PR3: proteinase 3.

## Data Availability

The de-identified data of the patients represented in the figures are available on request from the corresponding author.

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
