# Peer review of "Cocaine- and Levamisole-Induced Vasculitis: Defining the Spectrum of Autoimmune Manifestations"

_jcm, 2024, doi:10.3390/jcm13175116_

Round 1

Reviewer 1 Report

Comments and Suggestions for Authors

The authors present a detalied review on cocaine and levamisole induced vasculitis. The topic is interesting, incresingly important, with a very scarce literature data. The review is very good organized, there is a consistent systematic approch, overall manuscript language is clear, and easy to follow.

The figures are very informative, attractive and easy to understand. They point to the extensive experience of authors on this topic, adding additional value to the  review. I have only few suggestion for improvement.

1.  Please reconsider to shorten the first paragraph in the Introduction on AAV. I suggest more broaden Introduction on vasculitis (not only on AAV).

2. Lines 327-329: Add 1-2 sentences to define/describe LAC

3. The conclusion is too long. Please reconsider shortening and pointing out only most important aspects.

Comments on the Quality of English Language

Minor English editimg would be beneficial.

Author Response

Reviewer 1

============

The authors present a detalied review on cocaine and levamisole induced vasculitis. The topic is interesting, incresingly important, with a very scarce literature data. The review is very good organized, there is a consistent systematic approch, overall manuscript language is clear, and easy to follow.

The figures are very informative, attractive and easy to understand. They point to the extensive experience of authors on this topic, adding additional value to the  review. I have only few suggestion for improvement.

  1. Please reconsider to shorten the first paragraph in the Introduction on AAV. I suggest more broaden Introduction on vasculitis (not only on AAV).
  • Thank you for your insightful suggestion. We have followed your advice and revised the initial paragraph of the Introduction to be more concise, while also broadening the scope to include vasculitis in general, not just AAV.
  • You can find the changes in the main text at lines 29-66.
  1. Lines 327-329: Add 1-2 sentences to define/describe LAC
  • We would like to thank for this insightful comment. We have added the requested sentences to define/describe LAC in lines 1063 – 1065 and you can fint it below.
  • LAC vasculopathy/vasculitis is a complex systemic syndrome that primarily affects the skin and leukocytes, where the inflammatory vascular effects of cocaine and levamisole exposure converge
  1. The conclusion is too long. Please reconsider shortening and pointing out only most important aspects.
  • Thank you for your suggestion. We have reviewed the conclusion and have made efforts to shorten it, focusing on highlighting only the most important aspects, as per your advice. Lines 2375-2395

Minor English editimg would be beneficial.

  • Thank you for your suggestion. We took great care to proofread and edit the manuscript with the aim of improving its clarity and readability.

Reviewer 2 Report

Comments and Suggestions for Authors

This manuscript, “Cocaine and Levamisole-Induced Vasculitis: Defining the Spectrum of Autoimmune Manifestations”, presents a comprehensive and thorough review of the autoimmune manifestations associated with cocaine and levamisole use, particularly focusing on the spectrum of vasculitis and related disorders. The authors have successfully outlined the clinical features, pathogenesis, diagnostic challenges, and treatment options for these conditions. The review is well-structured and provides a valuable resource for clinicians and researchers alike.

Comments:

The manuscript is well-written, but there are occasional instances where the language could be more concise.

Ensure that all abbreviations are defined upon first use to avoid confusion.

This manuscript is a valuable addition to the literature on drug-induced vasculitis. With some revisions to improve clarity and focus and the inclusion of additional clinical guidance, it has the potential to serve as a key reference for clinicians and researchers in this field.

The section on treatment (Section 5) is underdeveloped. It lacks specific details on therapeutic strategies, particularly in cases where patients do not respond to drug cessation alone.

There is some redundancy in the discussion of the pathogenesis and clinical features of cocaine and levamisole-induced disorders. The manuscript could benefit from a more streamlined presentation, reducing overlap and ensuring that each section contributes distinct information.

Author Response

Reviewer 2

============

  • This manuscript, “Cocaine and Levamisole-Induced Vasculitis: Defining the Spectrum of Autoimmune Manifestations”, presents a comprehensive and thorough review of the autoimmune manifestations associated with cocaine and levamisole use, particularly focusing on the spectrum of vasculitis and related disorders. The authors have successfully outlined the clinical features, pathogenesis, diagnostic challenges, and treatment options for these conditions. The review is well-structured and provides a valuable resource for clinicians and researchers alike.

Comments:

The manuscript is well-written, but there are occasional instances where the language could be more concise. Ensure that all abbreviations are defined upon first use to avoid confusion. This manuscript is a valuable addition to the literature on drug-induced vasculitis. With some revisions to improve clarity and focus and the inclusion of additional clinical guidance, it has the potential to serve as a key reference for clinicians and researchers in this field.

  • Thank you for your kind words and for taking the time to review our manuscript. We appreciate that you found it suitable for publication with only minor revisions. We have carefully followed your suggestions to improve the manuscript, revising the English for conciseness and ensuring that all abbreviations are clearly defined upon their first use.

The section on treatment (Section 5) is underdeveloped. It lacks specific details on therapeutic strategies, particularly in cases where patients do not respond to drug cessation alone.

  • Thank you very much for your insightful feedback. We have carefully considered your suggestions and have made revisions accordingly. Specifically, we have expanded Section 5 to address the therapeutic strategies for cases where patients do not respond to drug cessation alone. You will find these improvements detailed between lines 2114-2379.

There is some redundancy in the discussion of the pathogenesis and clinical features of cocaine and levamisole-induced disorders. The manuscript could benefit from a more streamlined presentation, reducing overlap and ensuring that each section contributes distinct information.

  • Thank you for your valuable feedback regarding the redundancy in the discussion of pathogenesis and clinical features of cocaine and levamisole-induced disorders. We have thoroughly revised the manuscript to streamline the presentation, reducing overlap and ensuring that each section provides distinct and complementary information. Significant efforts were made in the section on pathogenesis, where we identified and removed several overlaps. Specifically, we have emphasized the three key pathogenic mechanisms (local tissue damage vs. autoimmunity/NETs vs. ANCA production) to provide a clearer and more structured presentation.

Furthermore, we have rearranged the subsections, switching the order of sections 2.1 and 2.2, and we moved the discussion on immune system activation/HNE from the "local tissue damage" section to the "ANCA production" section. These changes aim to improve clarity by better delineating the three mechanisms. We believe these revisions address the redundancy issue and enhance the overall clarity of the manuscript. Significant examples of these improvements can be found between lines 128-447.
